# New Atypical Antipsychotics in the Treatment of Schizophrenia and Depression

**DOI:** 10.3390/ijms231810624

**Published:** 2022-09-13

**Authors:** Jolanta Orzelska-Górka, Joanna Mikulska, Anna Wiszniewska, Grażyna Biała

**Affiliations:** Chair and Department of Pharmacology and Pharmacodynamics, Medical University of Lublin, Chodzki 4a, 20-093 Lublin, Poland

**Keywords:** schizophrenia, depression, aripiprazole, cariprazine, lurasidone, asenapine, brexpiprazole, lumateperone, pimavanserin

## Abstract

Schizophrenia and depression are heterogeneous disorders. The complex pathomechanism of the diseases imply that medication responses vary across patients. Many psychotropic drugs are available but achieving optimal therapeutic effect can be challenging. The evidence correlates well with clinical observations, suggesting that new atypical antipsychotic drugs are effective against negative and cognitive symptoms of schizophrenia, as well as against affective symptoms observed in depression. The purpose of this review presents the background and evidence for the use of the new second/third-generation antipsychotics (aripiprazole, cariprazine, lurasidone, asenapine, brexpiprazole, lumateperone, pimavanserin) in treatment of schizophrenia and depression. We have first provided a brief overview of the major neurobiological underpinnings of schizophrenia and depression. We then shortly discuss efficacy, safety and limitations of ongoing pharmacotherapy used in depression and schizophrenia. Mainly, we have focused this review on the therapeutic potential of new atypical antipsychotic drugs—currently existing—to be effective in psychotic, as well as in affective disorders.

## 1. Introduction

The first antipsychotic drug, chlorpromazine, was discovered in 1951. It was initially developed as an antihistamine to reduce intraoperative autoimmune stress [1]. Chlorpromazine represents the first-generation antipsychotics, also called typical antipsychotics/neuroleptics. In 1990, clozapine, the first atypical antipsychotic, was introduced into clinics. Typical agents are thought to work by antagonizing dopamine−2 (D_2_) receptors and to treat the positive symptoms of psychosis such as hallucination and delusion. Atypical antipsychotics treat both positive and negative symptoms of schizophrenia, the latter including decreased motivation and ability to feel pleasure, as well as social withdrawal. Thus, atypical antipsychotics are known to improve not only the core symptoms of schizophrenia but also the affective symptoms of depression that are frequently present in schizophrenic patients. Currently, the use of antipsychotic drugs, as monotherapy or as adjuncts to antidepressant and mood-stabilizing medicines, is also common practice for mood disorders that are not necessarily associated with psychosis [2].

Schizophrenia and depression—two severe mental disorders—are a major public health concern because they add to the global disease burden.

Schizophrenia is a chronic mental disorder affecting approximately 1% of the global population [3]. The diagnostic criteria of the disease include three main groups of symptoms: positive symptoms, negative symptoms and cognitive impairment [4]. Positive symptoms appear in patients in the form of hallucinations, delusions and disorders of thought processes often leading to aggressive and risky behaviour. A large group of patients also experience persistent negative symptoms (deficits), which include impaired speech, isolation, deprivation of emotion and pleasure feeling, and avolition. A separate group of symptoms is cognitive dysfunction, leading to deficits in verbal and working memory, prolonged information processing time, reasoning and attention disorders. It is estimated that negative symptoms affect 40% of patients, and marked cognitive impairment covers up to 80% [5]. 

Depression is one of the most common mental disorders with complex aetiology. It is a disease whose symptoms can be both recurrent and chronic. It can manifest as a gradual loss of enjoyment of life, depressive thoughts, anxiety, a sense of meaninglessness in life, sleep disturbances, decreased libido, and menstrual disorders. Despite much research into the treatment of depression, its aetiology is still not fully understood. The disease may be caused by genetic, epigenetic and environmental factors [6]. A better understanding of the pathophysiology of depression may significantly facilitate the development of new, more effective treatments.

The purpose of this review presents the background and evidence for use of the new second/third-generation antipsychotic in treatment of schizophrenia and depression. We have first provided a brief overview of the major neurobiological underpinnings of schizophrenia and depression. We then briefly discuss efficacy, safety and limitations of ongoing pharmacotherapy used in depression and schizophrenia. Mainly, we have focused this review on the therapeutic potential of new atypical antipsychotic drugs—currently existing—to be effective in psychotic, as well as in affective disorders.

## 2. Neurobiology of Schizophrenia

### 2.1. Dysregulation of the Dopaminergic System

The dopaminergic system is involved in motor performance, emotional behaviour and cognitive function. Impaired dopaminergic transmission leads to the development of mental illnesses such as schizophrenia and depression [3,7]. Post-mortem studies in patients with schizophrenia show increased levels of dopamine (DA) and its metabolites as well as an increased amount of DA receptors in the ventral striatum (Appendix A) [8,9]. Overstimulation of D_2_ receptors may be responsible for the positive symptoms of schizophrenia [9]. Moreover, it was found that levodopa (a DA precursor) and amphetamine (a DA releasing agent) exacerbate schizophrenic symptoms [10,11]. In addition, neuroimaging studies show increased DA release in the ventral striatum after amphetamine induction in patients with schizophrenia compared to a control group, which may indicate increased sensitivity of the dopaminergic system [7]. The appearance of negative symptoms is found with a decrease in dopaminergic activity of the mesocortical pathway and reduced stimulation of D_1_, D_3_ and D_4_ receptors in the prefrontal cortex [9].

### 2.2. Glutamatergic Transmission Disorder

It has been shown that abnormalities in the regulation of glutamatergic transmission may contribute indirectly to positive and negative symptoms and cognitive dysfunction [12]. The basis of the well-known glutamatergic hypothesis was the altered sensitivity of the N-methyl- D-aspartate receptor (NMDAR) located at Ɣ-aminobutyric acid (GABA) receptors in the cerebral cortex [13]. Secondary glutamatergic neurons, which directly stimulate dopaminergic neurons in the mesolimbic pathway, are inhibited [14]. Overactivity of the dopaminergic system can lead to the manifestation of positive symptoms [13]. NMDA receptor deficiency may also indirectly lead to inhibition of dopaminergic transmission in the mesocortical pathway, whose hypoactivity correlates with the onset of negative symptoms and cognitive dysfunction (Appendix A) [12]. Glutamatergic receptor antagonists—ketamine and phencyclidine—induce symptoms typical of schizophrenia in healthy individuals. After administration of the drug, DA release is amplified, more pronounced in schizophrenic patients, suggesting a relation between glutamatergic and dopaminergic transmission. Some postmortem studies have shown reduced expression of the GluN1 subunit of the NMDA receptor in patients in the prefrontal cortex. However, other postmortem studies of human brain tissue have revealed reduced signal transduction despite increased NMDA receptor expression. Nevertheless, the results of these studies implicate an underactivity of the glutamatergic system. The causes may be related to dysfunction of NMDA receptors as well as dysfunction of receptor modulators. Decreased levels of D-serine, an endogenous NMDA receptor agonist, and increased levels of natural antagonists such as kynurenic acid and N-acetylaspartylglutamic acid (NAAG), are detected in patient tissues postmortem. Increased concentrations of NAAG correlate with decreased concentrations of the antagonist-regulating enzyme, glutamate carboxypeptidase II (GCP-II). Genetic variants associated with glutamatergic transmission have also been identified. Analysis of DNA copy number variation (CNV) revealed de novo mutations in NMDAR genes and proteins affecting postsynaptic receptor density [12].

### 2.3. Increased Serotonergic Transmission in the Cortex

The result reveals the potential participation of serotonergic activity in the onset of psychosis. This may be due to excessive neurotransmitter release or increased expression of receptors for serotonin (5-hydroxytryptamine; 5-HT), with the 5HT_2A_ receptor playing a key role [13]. The data revealed that 5-HT receptors located on glutamatergic neurons in the cerebral cortex are crucial for the pathophysiology of some forms of psychosis such as auditory hallucinations and paranoid delusions (typical positive symptoms of schizophrenia). Hyperactivation of 5-HT_2A_ receptors could lead to glutamate release in the ventral tegmental area (VTA) and activation of the mesolimbic pathway, which results in excess DA in the ventral striatum (Appendix A) [13]. Lysergic acid diethylamide (LSD) and psylocybin (non-selective 5-HT_2A_ receptor agonists) are known to cause paranoia, distortions in perception and cognition that closely mimic some schizophrenic symptoms [15]. On the other hand, the blocking of 5-HT_2A_ receptors in the prefrontal cortex can increase the activity of the dopaminergic system in this brain structure. It may be important for the action of the atypical antipsychotic drugs and their possible effect on deficit symptoms and cognitive dysfunctions in schizophrenia [16]. Patients with Parkinson’s disease may suffer from secondary psychosis correlated with increased DA levels in dopaminergic pathways [17]. After administration of serotonin-2A (5HT_2A_) receptor antagonist (pimavanserin), with no effects on D_2_ receptors, suppression of the psychotic symptoms associated with Parkinson’s disease psychosis (PDP) was observed [17,18].

## 3. Neurobiology of Depression

### 3.1. The Monoaminergic Hypothesis

The main theory of the pathophysiology of depression is that the illness results from impaired monoaminergic transmission (including three monoamines: 5-HT, DA and noradrenaline (NE)). This hypothesis grew out of observations that antidepressant therapies raise neurotransmission tone depending on one or more of these neurotransmitters. In addition, the association of depression with neurodegenerative disease of the basal ganglia such as Parkinson’s and Huntington’s implicated DA [19]. Impaired monoaminergic transmission could be a consequence of monoamine depletion, impaired synthesis and regulation of the activity of monoamines (neurotransmitter reuptake by the specific transporter) and also of altered excitability/expression of the receptors. Moreover, there is functional connectivity of monoaminergic neurons—direct and indirect interconnections between 5-HT, NE and DA neurons are mediated through various receptor types which act on both autoreceptors and heteroreceptors. In particular, the impact of 5-HT systems on NE and DA neurotransmission were shown to be complex through 5-HT_2A_ and 5-HT_2C_ receptor-mediated mechanisms, respectively [20]. On the other hand, complex positive and negative influences of the NE system on 5-HT neurotransmission are mediated through α_1_- and α_2_-adrenergic receptors, respectively. Thus, the multimodal effect on central monoamine neurotransmission—the influence on reuptake transporters and on the different monoamine auto/hetreroreceptors—seem to improve the effectiveness of the therapy of resistant depression [2,19,21].

Postmortem and imaging studies indicate that the density of postsynaptic 5-HT_1A_ receptors is generally reduced in patients with depression. Moreover, the delayed onset of action characteristic of selective serotonin reuptake inhibitors (SSRIs) and selective serotonin-noradrenaline reuptake inhibitors (SNRIs) may reflect the time required to desensitize cell body 5-HT_1A_ autoreceptors. It is hypothesized that initially the 5-HT_1A_ autoreceptors compensate for inhibited 5-HT reuptake by decreasing the release of 5-HT from serotonergic neurons. As a result, the 5-HT transmission, especially at the nerve terminal level, remains unchanged. Chronic drug administration results in 5-HT_1A_ autoreceptor desensitization, which inactivates this negative feedback mechanism, thereby allowing marked increase in extracellular 5-HT and activation of postsynaptic 5-HT receptors [2,19,21]. 

Conversely, the 5-HT_2A_ receptors have both excitatory and inhibitory roles depending on brain region and appear to be an important site of action of atypical antipsychotics. Interestingly, patients with depression who committed suicide show increased expression of 5-HT_2A_ receptors in the prefrontal cortex and, in contrast, lower expression and reduced 5-HT_2A_ receptor binding affinity in the hippocampus compared with control (Appendix A). Evidence suggests that antagonism of the 5-HT_2A_ receptor potentiates NE release under SSRI treatment [19,22].

### 3.2. Neurogenesis and Neuroplasticity

The human brain is characterized by remarkable plasticity. One of the main factors supporting the survival of neurons and stimulating their growth and differentiation is brain-derived neurotrophic factor (BDNF). Preclinical studies in rats [23] have shown that the antidepressant effect of fluoxetine is associated with increased neurogenesis of the hippocampus. BDNF may play a key role in this process [24]. Hall et al. [25] demonstrated an association between stressful early childhood events and BDNF polymorphisms. This increases the risk of developing depressive disorders later in life.

It is thought that BDNF may play an important role in the pathogenesis of depression. The BDNF receptor tyrosine kinase B (TrkB) pathway represents a possible target for new antidepressants. It is important for neuronal survival and maturation. Interestingly, a relationship between the BDNF-TrkB pathway and serotonergic transmission has also been observed. Depression is thought to be a disease in which two brain systems are dysfunctional, i.e., the brain reward system (VTA-ventral tegmental area-nucleus accumbens (NAc) and VTA-prefrontal cortex pathways) and hippocampus-hypothalamic-pituitary-adrenal (HPA) pathway (Appendix A). Intra-hippocampal administration of BDNF shows an antidepressant effect, but also a pro-depressive effect on the reward system. The effects of BDNF on these two pathways need to be further investigated [26]. There are studies indicating an important role for BDNF in the function of serotonergic neurons. Expression of BDNF and its receptor TrkB takes place in serotonergic neurons. BDNF increases survival and differentiation of serotonergic neurons. Administration of BDNF to the brain also leads to demand-dependent regulation of 5-HT reuptake. It is thought that an increase in extracellular 5-HT concentration may increase BDNF concentration. However, this hypothesis requires further research [27,28].

## 4. New Atypical Antipsychotics in the Treatment of Schizophrenia

Antipsychotics are a key part of the treatment of acute episodes of schizophrenia and maintenance pharmacotherapy. The first drugs antagonizing the D_2_ receptor confirmed the dopaminergic hypothesis and proved effective in the treatment of psychosis. This discovery initiated the introduction of subsequent drugs with different chemical structures but similar pharmacological effects. The disadvantage of the drugs is low selectivity to the location of action [26]. In addition to blocking receptors at the target site, they also antagonize receptors in the other dopaminergic pathways [9]. Due to its non-selectivity and high affinity, this generation is fraught with numerous side effects, which include significant extrapyramidal disturbances, hyperprolactinemia and cognitive decline [26]. Typical antipsychotics are also associated with an increase in deficit symptoms due to a blockade of DA receptors in the nigrostriatal pathway. Other important side effects include sedation due to histamine receptor blockade and cardiovascular disorders due to antagonism to the α_1_ adrenergic receptor [9].

The second generation includes ‘atypical’ neuroleptics such as clozapine and risperidone [26]. In addition to blocking D_2_ receptors in the mesolimbic pathway, they show antagonism towards 5HT_2A_ receptors, which contributes to a partial reduction of negative symptoms. Antagonism of 5-HT_2A_ receptors may induce antipsychotic effects by altering dopaminergic tone. Blockade of 5HT_2A_ receptors leads to decreased dopaminergic transmission in the mesolimbic pathway indirectly by the decrease in glutamate release in the VTA (compared to the Appendix A) [29]. The atypical neuroleptics have less pronounced side effects, which may be a result of a lower affinity for D_2_ receptors or a high degree of dissociation. It has also been suggested that atypical neuroleptics have a higher preference for mesolimbic than nigrostriatal pathway receptors, which may result in a lower frequency of extrapyramidal disorders. However, an increase in patient weight is observed with their use [9]. 

The newest drugs are sometimes referred to as third generation antipsychotics. They are characterized by an extended receptor profile, including DA and 5-HT receptor subtypes together with a significant partial agonism to D_2_/D_3_ and 5-HT_1A_ receptors. Meta-analysis of postmortem studies found an elevation in prefrontal 5-HT_1A_ receptors in schizophrenia vs healthy controls [30]. The 5HT_1A_ receptor is an autoreceptor and its stimulation leads to inhibition of 5-HT release and subsequent inhibition of DA release in the prefrontal cortex (a decrease of DA in the prefrontal cortex is responsible for negative symptoms of schizophrenia). It is hypothesised that 5HT_1A_ receptor partial agonism may increase DA levels in the prefrontal cortex (minor effect of atypical antipsychotics on negative symptoms) [20,30]. It was also suggested that 5HT_2C_ agonism may result in antipsychotic effects without induction of extrapyramidal symptoms [9]. Promising molecular targets also include 5HT_6_ and 5HT_7_ receptors [22]. 

Aripiprazole (approved in 2002) is referred to as a “dopamine stabilizer” due to its partial agonism towards the D_2_ receptor. Partial agonists have a concentration-dependent effect on the endogenous neurotransmitter. At high concentrations of DA, they can block its receptors in the mesolimbic pathway and have an antipsychotic effect. At low concentrations observed in the prefrontal cortex, they behave as agonists and stimulate DA receptors [9]. Aripiprazole’s intrinsic activity, which does not cause excessive blockade of dopamine D_2_ receptor-mediated signalling, may explain its clinical effectiveness and a favourable profile of safety and tolerability. Contrary to the conventional antipsychotics, which completely block the D_2_ receptor-mediated physiological response to DA, aripiprazole can work as a functional antagonist and as a functional agonist in areas of overactivity and underactivity, respectively. For patients with schizophrenia, the onset of symptoms appears to be associated with increased activity of mesolimbic dopaminergic neurons (i.e., onset of positive symptoms) as well as decreased mesocortical activity (i.e., onset of negative symptoms and cognitive impairment) [31]. 

The promising action of aripiprazole may lead to the development of new drugs with similar receptor activity, but potentially better tolerability and safety profile. Two new dopamine receptor partial agonists, brexpiprazole and cariprazine, are now available. Although the mechanisms of action are similar, the three agents differ in terms of their pharmacodynamic profiles [31,32]. 

Brexpiprazole (*Rexulti^®^*) is a new atypical neuroleptic discovered by Otsuka Pharmaceutical Co., Ltd. in 2007 [33]. The drug was approved by the Food and Drug Administration (FDA) in the United States (2015), in Australia and Canada (2017) and in the EU and Japan (2018) for the treatment of schizophrenia [31,33,34]. The drug is a 5-HT and DA modulator that is chemically, structurally and pharmacologically similar to aripiprazole [34]. It exhibits partial agonism to the D_2_/D_3_ and 5HT_1A_ receptors, antagonism to 5HT_2A_ and α-1B/2C receptors. It has about 50 times lower affinity for histamine-1 (H_1_) than for D_2_ and 5HT_1A_ [35] and very low affinity for M_1_ muscarinic receptors [33]. Lower intrinsic activity to the D_2_ receptor (similar potency, lower intrinsic activity) and the increased potency of antagonism towards 5HT_2A_ receptors is related to potential reductions in akathisia, insomnia, restlessness and nausea in patients, often reported after use of aripiprazole. Significantly lower affinity for H_1_ receptors may be associated with attenuation of sedative effects and low weight gain [35]. The potential ability to induce D2 receptor hypersensitivity was evaluated in rats. After three weeks of administration of haloperidol and brexpiprazole to rats, a low dose of apomorphine was administered. The study showed that brexpiprazole, in contrast to haloperidol, did not significantly increase stereotypic behaviour [31] and did not cause hypersensitivity after repeated administration [31,36]. Brexpiprazole in vivo and ex vivo studies showed affinity to 5-HT_6_ and 5-HT_7_ receptors, but in applied doses it had no clinically significant effect. The antagonistic effect on adrenergic receptors is not fully understood. Based on studies in genetically modified mice, it is postulated that α_1B_ receptor blockade affects the reward system and may have antipsychotic effects, while α_1A_ receptor blockade may affect blood pressure and have general stimulant effects. Antagonism of α_2C_ receptor may have a pro-cognitive and antidepressant effect. However, further studies in this area are needed [31,33,34].

In Phase III double-blind, randomized, placebo-controlled clinical trials, brexpiprazole was effective in acute exacerbations and maintenance treatment and showed a good safety and tolerability profile. The study included adults aged 18–65 years with a current diagnosis of schizophrenia [36,37]. Safety and tolerability assessment showed that the drug was well tolerated in all patient groups. The incidence of akathisia was the only symptom reported twice as often in the brexpiprazole treatment groups and was 6%. Weight gain compared with placebo was moderate [35], averaged 1.2 kg in the first six weeks and 3.2 kg at the end, whereas placebo averaged 0.2 kg and 2 kg, respectively. This effect was considered to be beneficial compared with treatment with other antipsychotics such as quetiapine, risperidone or olanzapine (average weight gain over 24 weeks was 3.7 kg, 3.7 kg and 4.6 kg, respectively) [38]. The incidence of drowsiness and sedation in patients was at placebo levels. Extrapyramidal side effects occurred in 6.2% of patients receiving placebo and 10.7% receiving brexpiprazole. Changes in prolactin levels were minimal and hyperprolactinaemia was reported by less than 1% of patients [35]. Generally, brexpiprazole represents a promising new drug in the pharmacotherapy of schizophrenia for both acute exacerbation and maintenance treatment. Similar efficiency to currently used antipsychotics and potentially better tolerability and safety profile position it high among schizophrenia treatments.

A potential target of new atypical neuroleptics is the D_3_ receptor located in regions of the limbic system. An important role in the regulation of reward system, emotion, motivation and attention was indicated. Observations suggest that blockade of D_3_ receptors inhibits DA release in the prefrontal cortex [39]. In 2015, cariprazine, a piperazine derivative, was approved for the treatment of schizophrenia and manic or mixed episodes associated with bipolar disorder (BD). Cariprazine is a partial agonist at the dopamine D_2_ and D_3_ and the serotonin 5HT_1A_ receptors [18] and exhibits antagonism towards the H_1_ receptor. It has no affinity for muscarinic receptors. Studies demonstrate its efficacy in the treatment of schizophrenia. Due to its mechanism of action, cariprazine does not cause intractable sleep disturbances or significant weight gain. It has marginal effects on metabolic processes. However, a higher incidence of akathisia was found compared to aripiprazole and brexpiprazole [32].

Lurasidone is an atypical antipsychotic drug approved by the FDA in 2010 for the treatment of schizophrenia and depressive symptoms in BD. The drug was initially approved for the treatment of adults and has been approved for paediatric patients since March 2019. Off-label indications include the treatment of bipolar mania and aggression on the autism spectrum. The recommended dose in the pharmacotherapy of schizophrenia is 40–80 mg per day [40]. Lurasidone is one of the few antipsychotics to receive a pregnancy category B. Studies in rats and rabbits have shown no teratogenic effects on the foetus [41].

Lurasidone is a full D_2_, 5HT_2A_, 5HT_7_ receptor antagonist and a partial 5HT_1A_ receptor agonist. It shows an affinity for adrenergic receptors—higher for α_2c_ and slightly lower for α_1_ and α_2A_, and binds weakly to D_1_, 5HT_2c_, histamine and muscarinic receptors. Blockade of the D_2_ receptor is probably responsible for the antipsychotic effect. Strong antagonism to 5HT_7_ receptors is associated with mood regulation and pro-cognitive effects [40,41].

A meta-analysis of eight randomized, double-blind, placebo-controlled clinical trials proved the efficacy of lurasidone on positive and negative symptoms as evaluated by the PANSS (Positive and Negative Syndrome Scale) score. Safety and tolerability analysis showed no clinically significant difference in weight gain between the drug and placebo. There was no significant effect on metabolic parameters such as triglycerides, total cholesterol, HDL, LDL, changes in fasting glucose and prolactin levels. The most common treatment-related adverse effects included akathisia, drowsiness and sedation, nausea, vomiting and dystonia [42].

Efficacy in reducing symptoms and preventing relapse is estimated to be at the level of currently available drugs, but the low risk of metabolic side effects and weight gain is important to note. The moderate effect on prolactin levels and extrapyramidal disorders is also important. Depending on individual preference, response to treatment and risk analysis, lurasidone may be a worthy treatment option for schizophrenia in selected patient groups [43]. Lurasidone is only one drug among three novel antipsychotics (lurasidone, brexpiprazole and cariprazine) which may provide a useful therapeutic option for patients with depressive symptoms associated with schizophrenia [44]. The relevance of this feature stems from the fact that depressed mood is an established risk factor for suicide in schizophrenia. Thus, further studies on the potential antisuicidal effects of lurasidone are warranted [40]. 

Asenapine was approved in 2009 by the FDA for the treatment of schizophrenia and BD in adults, and subsequently for adjunctive treatment of the same disease entities and monotherapy of BD in the paediatric population [45]. It is classified as a receptor-targeted multi-acting antipsychotic. Asenapine possesses a high affinity for the serotonin 5-HT_2A_ receptor and to a lesser extent for the dopamine D_2_ receptor. The antagonist effect on D_2_ and 5-HT_2A_ receptors is considered clinically relevant. It is an antagonist of 5-HT_2C_, 5-HT_7,_ H_1_, α_1_ and α_2_-receptors. In contrast, it has no affinity for muscarinic receptors. This pharmacological profile provides the advantage that asenapine has a low risk of anticholinergic side effects (constipation and dry mouth, among others) [9,46]. 

Asenapine shows poor bioavailability < 2% after oral administration due to strong hepatic metabolism and for this reason is not used orally. The bioavailability of the sublingual form is 35% (at a dose of 5 mg). The drug is used twice daily [47]. In 2019, the FDA approved a transdermal form of the drug. This form is used once daily as opposed to the sublingual form [45]. A transdermal patch, HP-3070, has proved to be an interesting form of asenapine, with studies showing analogous efficacy and improved safety. Side effects affect fewer patients and are described as mild. In addition, use of the patch reduces the risk of oral hypersensitivity and taste disorders. This form may increase the efficacy of pharmacotherapy by reducing the number of patients discontinuing it due to bothersome side effects [48]. Asenapine is not contraindicated in patients with renal failure of varying severity or patients with mild to moderate liver failure [47]. The safety and efficacy of asenapine was confirmed in a double-blind, randomised clinical trial with placebo and olanzapine as control [49]. A study in a paediatric population involving children aged 10–17 showed efficacy and safety of the drug (dosage of 1–10 mg twice daily for up to 12 days) at the same level as in adults [50]. 

Lumateperone (ITI-007) was approved by the FDA in December 2019 for the treatment of schizophrenia in adults [18,44]. It has several unique features among antipsychotic medications. It acts synergistically through multiple systems (serotonergic, dopaminergic and glutamatergic), thus representing a unique approach for therapeutic management of a range of psychiatric disorders. Based on the pharmacodynamic profile of this drug and preliminary clinical data, depressive and negative symptoms as well as cognition may be specific domains of lumateperone action. It is a postsynaptic D_2_ receptor full antagonist that achieves maximal antipsychotic effect at only 39% receptor occupancy, but it acts as a partial agonist at the presynaptic D_2_ receptor. Additionally, lumateperone is a serotonin transporter (SERT) inhibitor with antagonist activity at serotonin 5-HT_2A_ receptors with affinity that is 60-fold higher than D_2_ receptors. Lumateperone is also a D_1_ receptor agonist and will indirectly increase phosphorylation of the NMDA receptor GluN2B. The drug has a therapeutic window, and doses higher or lower than the recommended 42 mg per day are ineffective. Lumateperone can be characterized by high efficacy in reducing psychosis in acute schizophrenia but with significantly fewer adverse effects [18,44]. 

Pimavanserin, an inverse agonist of the serotonin 5HT_2A_ receptor and, to a lower extent 5HT_2C_ receptor [51], was approved in 2016 for the treatment of hallucinations and delusions that occur in up to 50% of patients with Parkinson’s disease [52]. Since pimavanserin does not bind at all to dopamine D_2_ receptors, it is the first non-dopaminergic antipsychotic since discovery of the first DA antagonist, chlorpromazine. Pimavanserin is an ideal pharmacotherapy for Parkinson’ disease psychosis because it can treat the psychotic symptoms without exacerbating the motor symptoms of Parkinson’s disease. Whereas, all current antipsychotic drugs, due to reducing dopamine neurotransmission, cause significant adverse effects and worsen efficacy of standard anti-parkinsonian drugs [18,53]. In addition, a pilot study demonstrated marked response to pimavanserin of refractory positive symptoms in clozapine-nonresponsive patients [53]. Furthermore, pimavanserin is a promising drug for dementia-related psychosis [18].

To this day, the major pharmacological treatment strategy for schizophrenia is based on direct modulation of D_2_ receptor activity. Typical agents are thought to work by inhibiting D_2_ receptors to treat the positive symptoms of psychosis while atypical antipsychotics to treat both positive and negative symptoms, the latter including decreased motivation and ability to feel pleasure, as well as social withdrawal via inhibition of D_2_, α_2_ and 5-HT_2A_ receptors. While both classes of medications can be especially helpful in the management of psychosis, the side effects present a wide range of complications for the patient. In light of these findings, there has been a shift to focus on other components (i.e., cognitive and negative symptoms) of schizophrenia rather than just the antipsychotic responsive positive symptoms that primarily engage the dopaminergic system.

## 5. New Atypical Antipsychotics in the Treatment of Depression

Research over the second half of the 20th century was strongly influenced by the discovery that agents that alter monoamine metabolism, particularly that of 5-HT and NE, relieved depressive symptoms. Those agents, monoamine oxidase inhibitors (MAOIs) and tricyclic antidepressants (TCAs) are nonspecific, and hence their therapeutic benefits are associated with substantial side effects. Newer, more targeted agents such as SSRIs and noradrenaline reuptake inhibitors (NRIs) are effective in relieving symptoms in a significant percentage of patients. Most commonly, SSRIs are chosen. SNRIs are the next group of first choice drugs. Inhibition of 5-HT reuptake leads to an increased concentration of this neurotransmitter in the synaptic gap, which enables postsynaptic stimulation of 5-HT receptors on the surface of nerve cells. These drugs are well tolerated by patients, do not impair cognitive function and are not cardiotoxic. The disadvantage of SSRIs is that their effect takes some time to manifest itself; it must take several weeks before the therapeutic effect appears. TCAs are the second-line drugs for the treatment of depression. They are highly effective in treating depressive symptoms but are not first-line drugs due to their serious side effects. They can be followed by aggression, drowsiness, attention deficit disorder, paraesthesia, weight gain, QT prolongation, constipation and nausea. In depression accompanied by anxiety disorders, it is recommended to use drugs that are simultaneously effective in generalized anxiety. These include SSRIs, venlafaxine and duloxetine. In case of depression with insomnia, trazodone, mirtazapine and agomelatine are used [54].

Patients with treatment-resistant depression are those with major depressive disorder (MDD) that has not responded adequately to treatment. Moreover, one-third of major depressive episodes are held to contain mixed components. The most frequent manifestations of mixed depression are irritability, distractibility and psychomotor agitation. Mixed depression often accompanies risky behaviour including impulsive suicide attempts. The early detection and treatment of these unstable conditions is therefore necessary [55]. Pharmacological treatment options include switching to a different antidepressant, the addition of another antidepressant of a different class, or use of an augmenting agent, such as anticonvulsants, lithium or atypical antipsychotics. Some atypical neuroleptics—olanzapine and quetiapine—have already clinically proven efficacy by several meta-analyses [56,57,58,59]. However, the use of olanzapine and quetiapine is associated with weight gain, and akathisia, parkinsonism or insomnia problems may also occur. Thus, a hope for patients with treatment-resistant depression is a group of newer atypical neuroleptics. Due to their unique mechanism of action, the drugs are safer for the patient and their use is associated with a lower risk of motor side effects. 

Current scientific data on the use of new neuroleptics include their indications in schizophrenia, but also in depression and bipolar affective disorder (BD). This review is devoted to the use of atypical neuroleptics in schizophrenia, depression and depressive episodes in the course of BD. Although BD and depression belong to the separate diagnostic categories according to the International Classification of Diseases (ICD-10) and the American Diagnostic and Statistical Manual (DSM-5), many concepts of pathogenesis of depression are extrapolated to BD [60]. Hence, there are some similarities in the pharmacotherapy of both disorders and in the indications of some neuroleptics we can find both depression and BD (Table 1).

It should also be noted that depressive symptoms are recognized as an important and distinct symptom domain in schizophrenia [44]. New atypical neuroleptics—aripiprazole, asenapine, lurasidone and brexpiprazole—are currently licensed for the treatment of MDD or depressive symptoms in the course of BD.

Aripiprazole is approved by FDA for the adjunctive treatment of MDD. It is a good choice in elderly patients (>65 years of age) because it does not impair cognitive function. Studies in adult patients aged 18–65 years have shown its efficacy in combination with SSRIs and SNRIs [57]. 44% of subjects experienced remission of the disorder, while none of the patients experienced remission after treatment with an SNRIs. Aripiprazole is also effective in the pharmacotherapy of depression in people over 60 years of age. However, aripiprazole use was associated with a risk of akathisia and parkinsonism [61]. The akathisia was usually mild and manageable by dose adjustment. There was no significant weight gain or increased cardiometabolic risk after taking aripiprazole. The majority of studies conducted used 1–3 mg as the starting dose and 5–10 mg as the target dose. Some studies indicate racial differences in the selection of the effective dose. These may be due to genetic differences in cytochrome P 450 polymorphisms [62].

Lurasidone is considered to be a new atypical neuroleptic with antidepressant activity. The activity on the 5-HT_7_, 5-HT_1A_ and a_2C_-adrenergic receptors is hypothesized to enhance cognition, and the 5-HT_7_ receptor is being studied for its potential role in mood regulation and sensory processing. Lurasidone has high affinity for the serotonergic receptors 5-HT_7_ and 5-HT_1A_. A slight effect on 5-HT_2C_ receptors is associated with a relatively small weight gain. Lurasidone was approved by FDA for the treatment of depressive episodes in the course of BD in 2013. In BD, broad dosage ranges (20–120 mg/day) were found to be effective [40]. In paediatric BD, a difficult to treat population, lurasidone monotherapy at 20–80 mg daily doses was shown to be an effective and well-tolerated treatment for acute depression. Youth aged 10 to 17 with BD depression receiving 20–80 mg of lurasidone monotherapy showed significant improvements in depression, anxiety, and quality of life compared to placebo. Most commonly reported adverse events included nausea and sedation, with comparable weight change between lurasidone and placebo-treated groups [63]. As suggested in the placebo controlled, double-blind RCT encompassing depressed patients with BD, lurasidone exhibits significant anxiolytic properties. Noticeable improvement in quality of life was also observed in the lurasidone sample. The authors set the starting dose of the drug at 20 mg/day [64]. There is also evidence suggested that adjunctive lurasidone might be more effective in preventing depressive rather than manic episodes in maintenance treatment of BD. In another placebo controlled RCT Loebel et al. [65] examined the effects of lurasidone adjunctive therapy with either lithium or valproate in patients with BD. As compared to the placebo group, patients receiving lurasidone at the mean daily dose of 31.8 mg (range: 20–60 mg) or 82.0 mg (range: 80–120 mg) had greater improvements in depressive symptoms starting from week three through week six. Additionally, there was a significant reduction in anxiety symptoms, and improvement of patient-rated functional impairment and quality of life. 

Another atypical neuroleptic with potential antidepressant properties is asenapine. Like lurasidone, asenapine is a potent 5-HT_7_ receptor antagonist, and there have been studies evaluating its efficacy in patients with BD [66]. The treatment of BD can be problematic as the use of antidepressants can lead to destabilization of the illness. Asenapine also acts on other 5-HT receptors as well as DA receptors. It shows strong antagonistic effects against dopamine D_2_ receptors, which are responsible for potential antidepressant properties, but also 5-HT_2C_, 5-HT_2A_, 5-HT_2B_, and 5-HT_6_ receptors. The efficacy of asenapine in the treatment of depressive symptoms in BD has been confirmed in clinical studies. Its efficacy was compared with another antipsychotic drug (olanzapine). Asenapine had better efficacy in the PANSS [67].

Brexpiprazole was approved for the adjunctive treatment of MDD in 2015. Partial agonism to 5-HT1A receptors gives the drug antidepressant and anti-anxiety properties by stimulating hippocampal neurogenesis and increasing catecholamine output from the forebrain. Antagonism of 5-HT7 and 5-HT2A receptors is responsible for increasing 5-HT release from the prefrontal cortex and improving sleep architecture. A meta-analysis showed that brexpiprazole used at doses of 2 and 3 mg in combination with an antidepressant significantly improved the Montgomery–Asberg Depression Rating Scale (MADRS) scores compared with the antidepressant used with placebo. There was no improvement in efficacy with the 1 mg dose [57]. Longer term studies need to be conducted to confirm the superior efficacy of brexpiprazole in the treatment of MDD compared to other atypical neuroleptics as only data limited to 52 weeks are currently available. A direct comparison of aripiprazole with brexpiprazole will allow a better assessment of their efficacy and safety profile.

Recently, a potent and selective 5-HT2A and 5-HT2C modulator, pimavanserin, was developed as a new alternative for the treatment of psychosis. The development of pimavanserin as a drug with a unique pharmacological profile opened a new therapeutic avenue in the treatment of a few central disorders, including schizophrenia, dementia-related psychosis and also treatment-resistant depression. Pimavanserin has been studied for improving depression and associated symptoms in patients with MDD who have responded inadequately to one or more courses of serotonergic antidepressant. Furthermore, treatment with pimavanserin is associated with significantly greater improvement in specific symptoms associated with depression such as impaired sexual function, anxiety, sleepiness and irritability. These symptoms are typically unaffected or even worsened by commonly used serotonergic antidepressants. A recently completed phase 2 study showed that adjunctive use of pimavanserin may improve depression and associated symptoms in patients with MDD who have inadequately improved with one to two adequate courses of antidepressant treatments [68]. Adjunctive pimavanserin significantly improved sleep/wakefulness disturbance during treatment of MDD, an improvement that was associated with greater improvement in function [69]. Pimavanserin exhibits potent activity as a 5-HT_2A_ inverse agonist with some activity as a 5-HT_2C_ antagonist, but unlike most other antidepressants, pimavanserin exhibits no activity at adrenergic, dopamine, histamine, or muscarinic receptors. Sexual dysfunction as a side effect is often attributed to monoaminergic and dopaminergic activity of antidepressants [68]).

It is currently believed that only one-third of depressed patients respond to first-line treatment. In refractory depression the treatment of choice is to combine antidepressants with atypical neuroleptics. Usually, lower doses are used than in schizophrenia. Combination treatment is more effective than monotherapy. The use of atypical antipsychotics with additional antidepressant activity appears promising. The choice of treatment should be guided by efficacy, tolerability, side-effect profile, and presence of other psychiatric disorders. Furthermore, depressive symptoms are recognized as an important and distinct symptom domain in schizophrenia.

## 6. Conclusions

Schizophrenia and depression are heterogeneous disorders. The complex pathomechanism of the diseases implies that medication responses vary across patients. Many psychotropic drugs are available but achieving optimal therapeutic effect can be challenging. Thus, there are significant reasons to search for ways to improve effectiveness of current pharmacotherapy. 

The evidence correlates well with clinical observations, suggesting that new atypical antipsychotic drugs are effective against negative and cognitive symptoms of schizophrenia, as well as against affective symptoms observed in depression. Therefore, the development of new antipsychotic drugs with improved effectiveness, better impact on functional impairment, and less side effects represents a significant advance in the treatment of schizophrenia and depression.

## Figures and Tables

**Table 1 ijms-23-10624-t001:** The characteristics (mechanism of action and indications) of the antipsychotics reviewed.

Name od Drug	Aripiprazole	Brexpiprazole	Cariprazine	Lurasidone	Asenapine	Lumateperone	Pimavanserin
**Mechanism of action**	D_2_ partial agonist 5-HT_1A_ partial agonist 5-HT_2A_ antagonist	D_2_/D_3_ partial agonist 5-HT_1A_ partial agonist 5-HT_2A_ antagonist α_1B_/_2C_ antagonist low affinity for H_1_ and M_1_ receptors	D_2_/D_3_ partial agonist 5-HT_1A_ partial agonist low affinity for H_1_ and 5-HT_2A_ receptors lack of affinity for M receptors	D_2_, 5-HT_2A_ and 5-HT_7_ antagonist 5-HT_1A_ partial agonist an affinity for adrenergic receptors–higher for α_2C_ and slightly lower for α_1_ and α_2A_ very low affinity for D_1_, 5-HT_2C_, H_1_ and M_1_ receptors	D_2_, 5-HT_2A_, 5-HT_2C_ and 5-HT_7_ antagonist an affinity for adrenergic receptors –α_1_ and α_2_ lack of affinity for M receptors	full postsynaptic D_2_ antagonist D_2_ partial presynaptic agonist 5-HT_2A_ antagonist D_1_ agonist SERT inhibitor	5-HT_2A_ and 5-HT_2C_ inverse agonist lack of affinity for DA receptors
**Indications**	schizophrenia acute mania unipolar depression (adjunct)	schizophrenia unipolar depression (adjunct)	schizophrenia acute mania	schizophrenia depressive symptoms in BD (monotherapy and adjunct)	schizophrenia BD (acute mixed and manic episodes)	schizophrenia	Parkinson’s disease psychosis

## Data Availability

Not applicable.

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
