# Peer review of "New Atypical Antipsychotics in the Treatment of Schizophrenia and Depression"

_ijms, 2022, doi:10.3390/ijms231810624_

Round 1

Reviewer 1 Report

The manuscript “New atypical antipsychotics in the treatment of schizophrenia and depression” by Orzelska-Górka et al is a review about the use of seven new antipsychotics in schizophrenia and major depressive disorder.

The subject is interesting; however, a revision is required to address a number of issues.

In several instances, the seven antipsychotics are unevenly treated, with some of them discussed very thoroughly while others more superficially analysed. A more consistent approach is suggested.

At line 396 and following, some specific trials are mentioned while meta-analyses had been discussed in the previous sentences which probably included them. The reason why these trials were specifically reported should be detailed.

Data supporting asenapine use in schizophrenia should be reported in analogy with the other new antipsychotics treated in the review.

The classification of antipsychotics into generations is repeated through the manuscript.

References mentioned in the text are missing from the list: Stepnicki et al 2018; Stahl 2018; Bruijnzeel et al., 2015; Marder et al 2017.  Li et al at line 42 is not adequate as reference about schizophrenia incidence.

At line 240, a suitable reference in support of the different pharmacodynamic profiles is provided for brexpiprazole and ariprprazole, but a reference is missing for cariprazine, which is also mentioned in the sentence.

Lines 223-224: the meaning of dopaminergic tension is not clear. Likewise, the sentence at lines236-237 should be reformulated.

The section starting at line 424 presents evidence of lurasidone efficacy as an antidepressant agent based solely on studies in rodents. In a review focussed on clinical efficacy, these studies do not offer adequate support, thus they are not relevant for this manuscript and should be removed.

Brexpiprazole is misspelled in the section discussing its properties as antipsychotic agent.

The language is generally comprehensible, but a number of errors require attention.

Author Response

Thank you very much for Your recommendations. We have prepared a revised version of our manuscript, which has been modified in order to meet all the recommendations of the Reviewer. Indeed, the comments were very useful to help us improve the manuscript. We hope that after the modifications we have made, the manuscript has been significantly improved.

To be in accordance with the recommendations of Reviewer we have made the following changes:

  1. To be more consistent we removed some part of the descriptions of reviewed antipsychotics and in the others places we added more information on individuals agents.
  2. The sentences about specific trials were excluded. Instead of them, the sentences concerning the purpose of the work and the reason of selecting individual drugs were added.
  3. Data supporting asenapine use in schizophrenia have been reported in analogy with the other new antipsychotics treated in the review.
  4. We removed the information on classification of antipsychotics from some places of the manuscript to avoid the repetitions.
  5. We added missing references. Li et. al at line 42 was replaced with the correct reference.
  6. We have added a suitable reference for cariprazine in the sentence at line 240.
  7. The sentences at lines 223-224 and 236-237 were reformulated to be more clear.
  8. We removed evidence of lurasidone efficacy as an antidepressant agent based on studies in rodents. Instead of these sentences we added information on lurasidone’s efficacy as antidepressant agent based on clinical trials.
  9. We corrected the name of brexpiprazole in the section discussing its properties as antipsychotic agent.
  10. We reread the manuscript carefully trying to correct the linguistic mistakes.

Yours sincerely,

Authors

Reviewer 2 Report

In this review, the authors present the background and evidence for the use of the new second/third-generation antipsychotics (aripiprazole, cariprazine, lurasidone, asenapine, brexpiprazole, lumateperone, pimavanserin) in treatment of schizophrenia and depression. This review showed the therapeutic potential of new atypical antipsychotic drugs to be effective in psychotic, as well as in affective disorders. The authors have done a sound review. However, there are several minor comments.

1.       The neurotransmitter systems presumed to be involved in psychiatric disorders are complex. I would like to suggest that the authors draw a figure and/or table to describe the neurotransmitter systems associated with schizophrenia and depression.

2.       To make it easier for the reader to understand, I would like to suggest that the authors make a table to describe the characteristics of the new second/third-generation antipsychotics.

3.       Page 5 line 242, “Brexiprazole” is “Brexpiprazole”. The same misnomer is found elsewhere.

4.       Page 10 line 467, “Piwavanserin” is “Pimavanserin”.

Author Response

Thank you very much for Your recommendations. We have prepared a revised version of our manuscript, which has been modified in order to meet all the recommendations of the Reviewer. Indeed, the comments were very useful to help us improve the manuscript. We hope that after the modifications we have made, the manuscript has been significantly improved.

To be in accordance with the recommendations of Reviewer we have made the following changes:

  1. We prepared two figures to describe the neurotransmitter systems associated with schizophrenia and depression. We decided to draw two figures – the first one for schizophrenia and the second one for the depression. As the Reviewer wrote, the neurotransmitter systems presumed to be involved in psychiatric disorder are complex, thus to make the draws more legible, we have prepared separately for each disease. Nevertheless, we tried to underline the mechanisms of the development of these two disorders which are similar.
  2. We made a table to describe the characteristics of the new second/third- generation antipsychotics.
  3. We corrected the name of brexpiprazole.
  4. We corrected the name of pimavanserin.

Yours sincerely,

Authors

Round 2

Reviewer 1 Report

The authors have addressed all concerns.

Author Response

Thank you very much for Your recommendations and the acceptance for all our corrections.